# PAS: Plug-and-Play Prompt Augmentation System

## Abstract

In recent years, the rise of Large Language Models (LLMs) has spurred a growing demand for plug-and-play AI systems. Among the various AI techniques, prompt engineering stands out as particularly significant. However, users often face challenges in writing prompts due to the steep learning curve and significant time investment, and existing automatic prompt engineering (APE) models can be difficult to use. To address this issue, we propose PAS, an LLM-based plug-and-play APE system. PAS utilizes LLMs trained on high-quality, automatically generated prompt complementary datasets, resulting in exceptional performance. In comprehensive benchmarks, PAS achieves state-of-the-art (SOTA) results compared to previous APE models, with an average improvement of 6.09 points. Moreover, PAS is highly efficient, achieving SOTA performance with only 9000 data points. Additionally, PAS can autonomously generate prompt augmentation data without requiring additional human labor. Its flexibility also allows it to be compatible with all existing LLMs and applicable to a wide range of tasks. PAS excels in human evaluations, underscoring its suitability as a plug-in for users. This combination of high performance, efficiency, and flexibility makes PAS a valuable system for enhancing the usability and effectiveness of LLMs through automatic prompt engineering. The codebase is available at https://anonymous.4open.science/r/ape_gen-4214/.

## 1 Introduction

Large language models (LLMs) have demonstrated exceptional performance across diverse tasks spanning myriad domains Touvron et al. (2023); meta llama (2024); Bai et al. (2023). Variations in task contexts demand tailored response paradigms from large language models and the flexibility in adjusting response style is key to meeting the diverse needs. Nonetheless, pursuing a specific application or enhancing certain capabilities through model fine-tuning involves considerable data collection costs, computational resources, and the potential for a seesaw effect.

To address this issue, prompt engineering has emerged as a crucial approach among the various techniques for LLMs due to its extremely low cost and significant enhancement of LLM performance Sahoo et al. (2024). This technique leverages the inherent capabilities of LLMs to understand and generate human-like text, enabling them to perform a wide range of applications from natural language understanding and generation to specialized domains such as medical diagnosis and legal analysis Meskó (2023); Giray (2023). In prompt engineering techniques, few-shot learning Brown et al. (2020) stands out because it provides a small number of examples to guide the model, thereby enhancing its task-specific performance. Chain-of-Thought (CoT) Wei et al. (2022) prompting guides models to reason through problems step-by-step, improving logical consistency and accuracy. In-context learning Dong et al. (2022), by embedding relevant examples and instructions directly within the prompt, allows models to adapt dynamically to new tasks.

Despite the potential of existing methods, prompt engineering is not user-friendly and requires meticulous design. Crafting effective prompts demands a deep understanding of both the model and the task at hand. This process can be time-consuming and often involves extensive trial and error to optimize performance. To tackle those weak points, automatic prompt engineering (APE) is designed for easier prompt generation Zhou et al. (2022b); Pryzant et al. (2023b); Cheng et al. (2023). Although APE models can automatically enhance the prompt, they have to use massive amounts of

human-labeled data Cheng et al. (2023); Ouyang et al. (2022); Rafailov et al. (2024). Additionally, previous methods failed to construct a flexible, user-friendly and effective APE model Zhou et al. (2022a); Pryzant et al. (2023a). They face the following three key challenges:

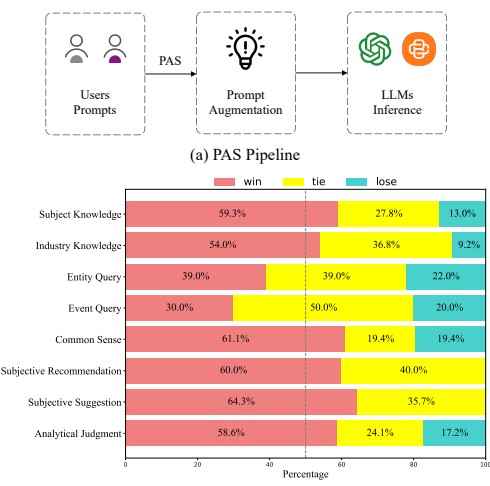

(a) PAS Pipeline

(b) Comparison of Win-Loss-Tie Outcomes With and Without PAS

Figure 1: We first present the pipeline of the PAS in (a). PAS takes user prompts, enhances them, and then inputs the augmented prompts into LLMs. As illustrated in (b), PAS significantly improves responses across all categories in human evaluation.

Figure 2: Case Study 1, Red text is the complementary prompt generated by PAS. We can see PAS can give complementary prompt to avoid logic traps.

**C1. Poor Effectiveness.** Although several works aim to automatically enhance the quality of prompts, they often rely on outdated evaluation metrics and do not consistently yield significant improvements across various benchmarks and models Cheng et al. (2023); Ouyang et al. (2022). Additionally, these models fail to include human evaluations, thus lacking valuable feedback from human users Cheng et al. (2023); Pryzant et al. (2023b).

**C2. Low Flexibility.** Previous works primarily focus on the performance of APE models, overlooking the importance of flexibility, specifically their model-agnostic and task-agnostic capabilities Pryzant et al. (2023b); Yang et al. (2023b). Low-flexibility APE systems can lead to computational waste and hinder their application on LLMs Pryzant et al. (2023b). Additionally, these low-flexibility systems have limited applicability across various scenarios, making them less versatile for diverse use cases and reducing their overall effectiveness in practical applications.

**C3. Limited Practicality** Among the previous methods, only BPO offers flexibility for prompt augmentation. However, BPO chose to rewrite the questions for prompt augmentation. this carries the risk of the new prompts deviating from the oiginal intent. Additionally, when dealing with long inputs, BPO is unable to control the generating time. It is also impossible for real-time display.

To address these issues, in this paper, we designed a plug-and-play system, called Prompt Augmentation System, PAS. The system employs a specialized model designed to generate prompts tailored to various tasks, thereby transforming the prompts into domain-specific expert prompts which are the inputs for LLM. The central idea behind PAS is to decouple the scene requirements from the model's capabilities, as shown in Figure 1(a), which is a two-stage alignment paradigm. In this two-stage approach, the prompt augmentation (PA) model is utilized to understand prompts and generate the specific constraints or hints for them, while the LLM focuses on instruction following and knowledge capabilities. The PA model and LLM collaborate to accomplish tasks. Prompt augmentation can be achieved in two ways: by entirely rewriting the prompt or by generating supplementary content to enhance the original prompt. BPO Cheng et al. (2023) adopted the rewriting approach; however, this method risks altering the original intent and results in unpredictable prompt generation

times, especially for lengthy inputs. To address these limitations, we opted to add complementary content following the original prompt. As illustrated in Figure 2, the PA model comprehends the original prompt, identifies potential logical traps, and generates supplementary content as a hint. The core contributions of this paper are summarized as follows:

- **New Perspective.** To the best of our knowledge, this work is the first to construct a curated prompt complementary dataset without human labor. Additionally, we are the first to utilize this dataset to train LLMs to construct the PAS to automatically complement user prompts instead of directly modifying them.

- **New Method for Auto Prompt Augmentation System.** We propose a new data collection method to construct a complementary prompt dataset for training a Prompt Augmentation model. Furthermore, we introduce a plug-and-play Prompt Augmentation System (PAS), which can be seamlessly integrated into any LLM to enhance model performance without additional training costs. PAS is designed to interpret prompts and generate specific constraints or hints to guide responses.

- **SOTA Performance. (1)** *SOTA Performance on Multiple Benchmarks.* Our PAS model achieves SOTA performance across multiple models and comprehensive benchmarks. It also outperforms the previous SOTA model, BPO, under identical experimental settings. Furthermore, it demonstrates superior performance on human evaluation metrics, as shown in Figure 1(b), highlighting the outstanding capabilities and potential applications of the PAS model. Figure 2 further illustrates that our model has good performance in real-world problems. **(2)** *High Flexibility.* Our PAS model can be plugged into any LLM, and is model and task-agnostic. It achieves SOTA performance across all models and tasks, demonstrating its exceptional flexibility. **(3)** *High Application Potential.* Since we choose to add complementary behind the prompt. Our PAS system excels in controlling generation time, enabling real-time word-by-word display, and efficiently handling prompt augmentation for long documents as well as retrieval-augmented generation (RAG), showcasing its substantial application potential.

## 2 RELATED WORK

### 2.1 AUTOMATIC PROMPT ENGINEERING

The effectiveness of large language models in various applications largely depends on the quality of the prompts used. There are already many designed prompts that can significantly enhance the performance of LLMs Kojima et al. (2022); Wei et al. (2022); Yao et al. (2024); Besta et al. (2024); Yang et al. (2024); Wang et al. (2023). However, these methods that rely on manual prompt engineering are far less scalable. In the field of mathematical logical reasoning for LLMs, the Chain of Thought and its derived strategies are widely popular due to their effectiveness. Zero-shot CoT Kojima et al. (2022) is adding a simple sentence like "Let's think step by step" at the end of questions to assist LLMs in generating reasoning steps. Instead of Zero-shot CoT, Manual-Cot Wei et al. (2022) provides reasoning steps as few shots. Self-Consistency further improves language models' reasoning performance by generating a diverse set of reasoning paths and choosing the most consistent answer in the final answer set. Tree of Thought (TOT) Yao et al. (2024) and GOT Besta et al. (2024) extend the reasoning pathway from linear to non-linear data structures by leveraging multiple LLM queries to elicit different plausible reasoning paths Yang et al. (2024). Buffer of Thought (BOT) Yang et al. (2024) designs a series of thought-template for tasks, and for each problem, it retrieve a relevant thought-template to prompt LLMs. PS prompting Wang et al. (2023) improves COT by encouraging LLMs to devise a plan before attempting to solve a problem.

Aforementioned prompting engineering strategies have been crafted by human expertise. To reduce manual effort, there is numerous recent work to explore how to conduct automated prompt engineering Zhang et al. (2022); Shum et al. (2023); Zhou et al. (2022a); Yang et al. (2023b); Pryzant et al. (2023a); Guo et al. (2023); Fernando et al. (2023). Auto-Cot Zhang et al. (2022) partitions questions of a given dataset into a few clusters and generates reasoning chains to construct demonstrations for each cluster for Few-shot COT. Automatic-COT Shum et al. (2023) creates rationale chains to augment exemplars and filters out incorrect ones by checking against the ground truth. Both of them improve the performance of vanilla COT Kojima et al. (2022); Wei et al. (2022). Unlike previ-

ous works, OPRO Yang et al. (2023b), APO Pryzant et al. (2023a), and APE Zhou et al. (2022a) provide an optimizer's perspective for automatically finding prompts. OPRO Yang et al. (2023b) leverages LLMs as optimizers, using the accuracy of training datasets—unavailable in real-world scenarios—as the objective value. APO Pryzant et al. (2023a) provides detailed guidance on prompt refinement at each step, based on the differences between responses and targets. Evoprompt Guo et al. (2023) and Promptbreeder Fernando et al. (2023) introduce evolutionary algorithms (EAs) into discrete prompt optimization for specific domains. Similar to evolutionary algorithms, they require evaluating the fitness of each individual prompt in the population, presenting significant challenges in practical applications. Additionally, exploring dozens of generations of prompts imposes a considerable burden.

## 2.2 Plug-and-Play Systems

Plug-and-play systems have garnered significant attention in recent years due to their modularity and ease of integration in various machine-learning workflows. These systems are designed to operate seamlessly with existing frameworks, allowing for quick and flexible augmentation of functionalities without the need for extensive reconfiguration Abdulrazak & Helal (2006); Zhang et al. (2021); Venkatakrishnan et al. (2013).

In image processing research, plug-and-play systems are commonly applied for its outstanding flexibility. Image reconstruction, denoising, deblurring, image enhancement, and super-resolution are all fields where plug-and-play systems are highly needed. By integrating various image processing modules into a unified framework, plug-and-play systems can flexibly combine different methods to achieve better image processing results. Moreover, this system allows for the easy addition or replacement of new processing modules without redesigning the entire algorithm, significantly improving the efficiency and effectiveness of image processing.

In the field of artificial intelligence, the rapid advancement of machine learning models has spurred a growing demand for plug-and-play systems. These systems enable seamless integration and adaptation of AI technologies across various applications. Al Ridhawi et al. (2020) have underscored the critical role of AI plug-and-play systems in enhancing scalability, flexibility, and usability in modern computational frameworks.

## 3 Method

In this section, we first summarize the collection and process of prompt data in section 3.1. Then we introduce the prompt complementary data generation pipeline in section 3.2 to automatically generate high-quality prompt complementary data. After the prompt dataset is generated, we give a comprehensive analysis of the dataset in section 3.3. At last, in section 3.4, we introduce utilizing the prompt augmentation dataset for LLM fine-tuning and then use the fine-tuned LLM to construct a plug-and-play APE system.

### 3.1 Prompts Data Collection

In this section, we introduce the prompt collection process. To generate high-quality (prompt, complementary prompt) pairs, we first need to select high-quality prompts. To achieve this, we use two curated datasets: the LMSYS-1M dataset Zheng et al. (2023) and the WildChat dataset Zhao et al. (2024). We use $P$ to denote the prompt dataset. As shown in Figure 3(a), our data selection process involves three main steps to ensure the quality and relevance of the data:

**Deduplication** Deduplication is applied to ensure a diverse and efficient dataset. All prompts from the LMSYS-1M and WildChat datasets are embedded using the SimCSE BGE model. Next, the HNSW clustering algorithm is employed to group these embeddings. From each cluster, a representative subset is selected to minimize redundancy. The resulting deduplicated prompt subset is denoted as $P^*$.

**Quality Selection** Quality filtering is performed to select high-quality data because such data can not only reduce computational costs but also enhance the model's performance. For quality selection, the BaiChuan 13b model Yang et al. (2023a) serves as the base model. We filter out low-quality

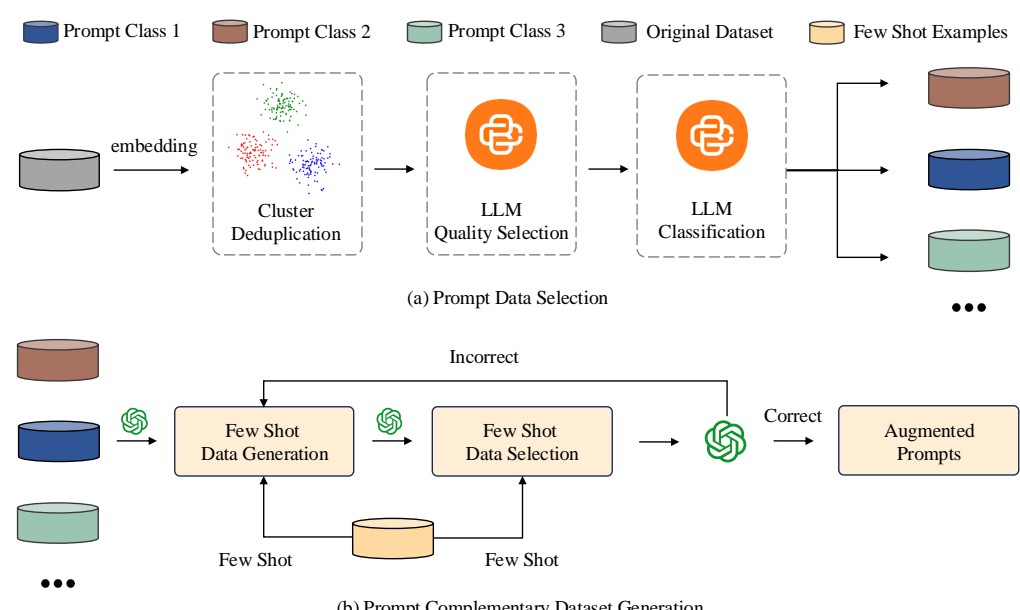

Figure 3: Pipeline for selecting prompt data and generating complementary prompt data.

entries using the formula below:

$$Q_{\text{score}}(p_i) = \text{BaiChuan 13b}(p_i)$$

$$P_{\text{filtered}} = \{p_i \in P^* \mid Q_{\text{score}}(p_i) \geq \tau\}$$

Here, $Q_{\text{score}}(p_i)$ represents the quality score assigned by the BaiChuan 13b model to prompt $p_i$, and $\tau$ denotes the quality threshold. By employing quality selection, we aim to enhance the overall quality of prompt data.

**Classification** To enhance performance, we first classify the prompts into multiple categories and then design specific few-shot learning examples tailored for each category, as detailed in Section 3.2. To achieve accurate classification, we fine-tune a BaiChuan 13b model Yang et al. (2023a) using 60,000 internally labeled classification data points from our collaborating organization. This results in a classification model capable of categorizing prompts into common categories such as Q&A and coding.

The deduplication, quality selection, and classification steps ensure data diversity, quality, and accurate categorization. Through these processes, we obtain approximately 9,000 high-quality classified data points, which are subsequently used to generate high-quality (prompt, complementary prompt) pairs.

## 3.2 PROMPTS COMPLEMENTARY DATASET

To generate a high-quality prompt complementary dataset, we designed an automated data generation pipeline based on Few-Shot Learning. The algorithm mainly consists of two phases: data generation and data selection with regeneration. We manually design 4-5 few-shot examples for each category, where we call it few-shot golden data $D_{\text{few-shot}} = \{(p_i, a_i)\}_{i=1}^{N}$. Then, we utilize the prompt dataset $P_{\text{golden}}$ from Section 3.1 to generate high-quality (prompt, complementary prompt) pairs utilizing few-shot learning.

**Data Generation** For each prompt $p_j \in P_{\text{golden}}$ in every category, we utilize the Few-Shot Learning method based on the prompt in Figure 10 to generate a corresponding complementary prompt $a_j$ based on the golden few-shot data $D_{\text{few-shot}}$. The generated prompt-complementary prompt pair $(p_j, a_j)$ is then added to $D_{\text{generated}}$.

**Data Selection and Regeneration** We observed that not all the generated complementary prompt data are of high quality or useful for the original prompt. To address this issue, we proposed a data selection and regeneration pipeline for high-quality complementary prompts.

For each generated prompt-answer pair $(p_j, a_j) \in D_{\text{generated}}$, we use Few-Shot Learning based on the prompt in Figure 11 to evaluate its correctness. If the evaluation result is incorrect, we remove the pair and use Few-Shot Learning in the data generation phase to regenerate the answer until the correct answer is generated. Finally, we add the correct prompt-answer pair back to $D_{\text{generated}}$.

Through this data selection and regeneration process, we can automatically generate a prompt complementary dataset while ensuring data quality. This process provides reliable data support for subsequent model training.

### 3.3 PROMPT COMPLEMENTARY DATASET

In this section, we present a comprehensive analysis of the generated prompt-complementary dataset. The dataset consists of approximately 9,000 high-quality (prompt, complementary prompt) pairs, as shown in Figure 12. Figure 12 provides an overview of the dataset distribution, indicating that it encompasses 14 categories, with each category containing roughly 500 data points. This extensive coverage of various prompt categories highlights the dataset's robust generalization capabilities. Additionally, the dataset includes a substantial amount of Coding and Q&A data, which are widely used functions, thereby justifying their significant representation.

Although the data distribution exhibits notable strengths, our automated process for generating complementary prompts, as described in Section 3.2, enables us to control the categories of generated data. This flexibility allows our method to accommodate both general-purpose models and domain-specific requirements, thereby improving prompt adaptability across different contexts. Tailoring data generation to specific domains ensures comprehensive training for diverse PAS tasks across multiple fields.

Overall, the extensive coverage of various prompt categories demonstrates the dataset's strong generalization capabilities, with a particular emphasis on critical functionalities. Furthermore, our method's ability to regulate dataset distribution facilitates the development of PAS systems across a wide range of applications.

### 3.4 AUTOMATIC PROMPT COMPLEMENTARY PLUG-AND-PLAY SYSTEM PAS

In this section, we first fine-tune LLMs using the prompt-complementary dataset introduced in Section 3.2 to perform prompt-complementary tasks. Due to the flexibility of LLMs in generating complementary prompts, these models can be seamlessly integrated into other generative LLMs. As a result, they can function as an automatic, plug-and-play system to enhance the performance of existing LLMs.

**Fine-tune LLMs for Prompt Complementary Models** By utilizing the high-quality generated dataset $D_{\text{generated}}$ in section 3.2, we can automatically fine-tune LLMs to have a prompt complementary ability. We use $M_p$ to denote an automatically prompt complementary model, and $M$ to denote an LLM, which can be written as the following mathematic formula:

$$M_p \leftarrow \text{SFT}(M; D_{\text{generated}})$$

We call this $M_p$ model PAS, a prompt complementary model which can serve as an automatic, plug-and-play system to enhance LLM performance.

**PAS Enhances LLMs Performance** For a prompt $p$ from the user, the complementary prompt $p_c$ generally serves as a supplementary hint or guidance to provide additional context, enhance clarity, or direct the model towards a desired response. The complementary prompt $p_c$ can be generated using the following formula:

$$p_c = M_p(p)$$

Table 1: Comparison of PAS, BPO and not using APE (baseline). We can see PAS significantly outperform the baseline with an average improvement of 8 points. Additionally, when compared to the previous state-of-the-art model, BPO, our model still exceeds it by an average of 6.09 points.

| Main Model | APE-model | Arena-hard | Alpaca-Eval 2.0 | Alpaca-Eval 2.0 (LC) | Average | ↑ |
|---|---|---|---|---|---|---|
| GPT-4-turbo-2024-04-09 |  | 76.60 | 46.12 | 55.02 | 59.25 | - |
| GPT-4-1106-preview |  | 74.80 | 50.00 | 50.00 | 58.27 | - |
| GPT-4-0613 |  | 37.9 | 15.80 | 30.20 | 27.97 | - |
| GPT-3.5-turbo-1106 | - | 18.90 | 9.20 | 19.30 | 15.80 | - |
| Qwen2-72b-Instruct |  | 48.10 | 31.70 | 39.24 | 39.68 | - |
| LLaMA-3-70b-Instruct |  | 41.10 | 33.18 | 34.42 | 36.23 | - |
| **Average** | - | 49.57 | 31.0 | 38.03 | 39.53 | - |
| GPT-4-turbo-2024-04-09 |  | 76.60 | 54.65 | 55.28 | 62.18 | +2.93 |
| GPT-4-1106-preview |  | 74.60 | 55.19 | 52.91 | 60.90 | +2.63 |
| GPT-4-0613 | BPO | 38.60 | 19.61 | 34.08 | 30.76 | +2.79 |
| GPT-3.5-turbo-1106 |  | 15.90 | 10.25 | 20.29 | 15.48 | -0.32 |
| Qwen2-72b-Instruct |  | 44.40 | 31.25 | 39.02 | 38.22 | -1.46 |
| LLaMA-3-70b-Instruct |  | 45.20 | 38.92 | 39.24 | 41.12 | +1.59 |
| **Average** | BPO | 49.22 | 34.98 | 40.14 | 41.44 | +1.91 |
| GPT-4-turbo-2024-04-09 |  | 76.90 | 65.31 | 56.54 | 66.62 | +7.37 |
| GPT-4-1106-preview |  | 78.80 | 65.92 | 53.63 | 66.12 | +7.85 |
| GPT-4-0613 | PAS | 43.90 | 34.06 | 40.33 | 39.43 | +11.46 |
| GPT-3.5-turbo-1106 |  | 22.10 | 15.82 | 23.31 | 20.41 | +4.61 |
| Qwen2-72b-Instruct |  | 52.20 | 45.53 | 44.31 | 47.35 | +7.67 |
| LLaMA-3-70b-Instruct |  | 50.30 | 45.01 | 40.52 | 45.28 | +9.05 |
| **Average** | PAS | 54.03 | 45.37 | 43.20 | 47.53 | +8.00 |

The enhanced response $r_e$ is then given by:

$$r_e = \text{LLMs}(\text{cat}(p, p_c))$$

By generating a complementary prompt, the PAS can improve the user's prompt without altering the original input. As demonstrated in Section 4, PAS effectively enhances the performance of LLMs.

**Plug-and-Play LLMs Promoting System** PAS can be plugged into any other LLMs available via public APIs OpenAI (2023) and can also be integrated into models with open parameters Bai et al. (2023); Touvron et al. (2023). This flexibility allows for a wide range of applications and improvements across different platforms and systems.

The primary advantage of such a system is its ability to seamlessly enhance the capabilities of existing LLMs without the need for extensive retraining or modification. By simply augmenting the input prompts, PAS leverages the strengths of the underlying models while providing a mechanism to improve their performance. Moreover, it can alleviate the trade-off problem commonly encountered during fine-tuning, where optimizing for one task often results in a decline in performance for other tasks. We will further discuss these advantages of our plug-and-play system in section D.

## 4 EXPERIMENTS

In this section, we first introduce the experimental setups. We then aim to answer the following questions to verify the effectiveness, efficiency, and robustness of our proposed PAS: **Q1**: Can our PAS achieve SOTA performance compared to previous SOTA methods? **Q2**: Can our PAS outperform the previous SOTA model with the same base model? **Q3**: How efficient and flexible is our model compared to previous APE models? **Q4**: Can PAS achieve SOTA performance in human evaluation, making it a user friendly system? **Q5**: Do we need data quality selection and regenerate module in our data generation pipeline?

### 4.1 EXPERIMENTS SETTING

**Datasets.** We followed the steps in section 3 and generated a dataset of 9000 high-quality pairs (prompt, complementary prompt).

**Models.** For PAS models, we select several smaller models to efficiently train a PAS model. We select Qwen2-7b-Instruct Bai et al. (2023), LLaMA-2-7b-Instruct Touvron et al. (2023) as base

model. Then we utilize our trained PAS models to some massive SOTA models, i.e. GPT-4-turbo-2024-04-09, GPT-4-1106-preview, GPT-4-0613, GPT-3.5-turbo-1106 OpenAI (2023), Qwen2-72b-Instruct Bai et al. (2023), and LLaMA-3-70b-Instruct Touvron et al. (2023); meta llama (2024).

**Baselines.** We compare the performance of PAS with models without PAS. Additionally, we compare the performance of PAS with the previous SOTA automatic prompt engineering method BPO Cheng et al. (2023) to demonstrate the effectiveness of PAS.

**Evaluation.** To evaluate the effectiveness of our PAS model, we used three comprehensive benchmarks Arena-hard, Alpaca-Eval 2.0 and Alpaca-Eval 2.0 (LC) to thoroughly assess the model's performance.

## 4.2 MAIN EXPERIMENTS

To address **Q1**, we used Qwen2-7B-Instruct as the base model due to its outstanding performance. We subsequently used the prompt complementary data to train a PAS model and compared it to both the baseline model without the APE model and the previous state-of-the-art (SOTA) APE model, BPO Cheng et al. (2023). We integrated our model into multiple commonly used LLMs, including GPT-4-turbo-2024-04-09, GPT-4-1106-preview, GPT-4-0613, GPT-3.5-turbo-1106 OpenAI (2023), Qwen2-72b-Instruct Bai et al. (2023), and LLaMA-3-70b-Instruct Touvron et al. (2023); meta llama (2024).

The results in Table 1 clearly illustrate the effectiveness of our PAS method across different models. Compared to the baseline without using APE, PAS shows significant improvements in all metrics, resulting in an average improvement of 8 points, demonstrating the benefits of incorporating prompt complementary data. For instance, in the case of GPT-4-0613, PAS improves the average score by 11.46 points compared to the baseline, highlighting its substantial impact.

Moreover, when compared to the previous state-of-the-art model BPO, our model significantly outperforms it, resulting in an average improvement of 6.09 points. Each model achieves more than a 4-point average improvement across all six base models compared to BPO, with a notable increase of 9.13 points for Qwen2-72b-Instruct, indicating a substantial improvement.

BPO is unstable and performs worse than the baseline in some cases, such as GPT-3.5-turbo-1106 and Qwen2-72b-Instruct, indicating that the previous SOTA model cannot consistently outperform the baseline. Considering our model exceeds the baseline by 8.00 points and BPO by 6.09 points, it further demonstrates the effectiveness and robustness of our PAS model.

Overall, our PAS method not only outperforms the baseline but also consistently surpasses the previous SOTA model BPO, establishing its robustness and effectiveness as a fine-tuning strategy for enhancing prompt-based learning systems. This consistent performance across various LLMs underscores the robustness of PAS and its potential to set new benchmarks in the field.

Table 2: Comparison of PAS and BPO using the same base model, LLaMA-2-7b-Instruct. The results demonstrate that PAS outperforms the BPO model consistently across all LLMs when using the same base model, LLaMA-2-7b-Instruct.

| Main Model | Method | Arena-hard | Alpaca-Eval 2.0 | Alpaca-Eval 2.0 (LC) | Average | ↑ |
|---|---|---|---|---|---|---|
| GPT-4-turbo-2024-04-09 | | 76.60 | 54.65 | 55.28 | 62.18 | - |
| GPT-4-1106-preview | | 74.60 | 55.19 | 52.91 | 60.90 | - |
| GPT-4-0613 | BPO | 38.60 | 19.61 | 34.08 | 30.76 | - |
| GPT-3.5-turbo-1106 | | 15.90 | 10.25 | 20.29 | 15.48 | - |
| Qwen2-72b-Instruct | | 44.40 | 31.25 | 39.02 | 38.22 | - |
| LLaMA-3-70b-Instruct | | 45.20 | 38.92 | 39.24 | 41.12 | - |
| **Average** | BPO | 49.22 | 34.98 | 40.14 | 41.44 | - |
| GPT-4-turbo-2024-04-09 | | 73.54 | 62.58 | 54.03 | 63.38 | +1.20 |
| GPT-4-1106-preview | | 75.52 | 64.06 | 53.07 | 64.22 | +3.32 |
| GPT-4-0613 | PAS | 40.13 | 33.11 | 36.70 | 36.65 | +5.89 |
| GPT-3.5-turbo-1106 | | 18.02 | 16.18 | 23.67 | 19.29 | +3.81 |
| Qwen2-72b-Instruct | | 47.91 | 40.59 | 39.99 | 42.83 | +4.61 |
| LLaMA-3-70b-Instruct | | 46.30 | 43.17 | 38.77 | 42.74 | +1.62 |
| **Average** | PAS | 50.24 | 43.28 | 41.04 | 44.85 | +3.41 |

## 4.3 Effectiveness of PAS

To address **Q2**, we fix the base model and compare our PAS method with the previous BPO Cheng et al. (2023). We use LLaMA-2-7b-Instruct, the same base model as BPO, and utilize the generated complementary prompt data to fine-tune LLaMA-2-7b-Instruct. We compare our model performance with BPO.

The results in Table 2 clearly demonstrate the effectiveness of our PAS method across different models. Notably, PAS exhibits a marked improvement in performance metrics compared to BPO, exceeding the baseline by 3.41 points on average. This is particularly evident in models like GPT-4-0613, where the average score improvement is as high as 5.89 points. Even in cases where the improvement is smaller, such as Llama3-70b-Instruct and GPT-4-turbo-2024-04-09, PAS still manages to outperform BPO for more than 1 point, indicating its robustness and consistency.

Overall, our PAS method consistently improves model performance across various evaluation models and settings, establishing its effectiveness and robustness as a fine-tuning strategy for enhancing prompt-based learning systems.

## 4.4 Human Evaluation

To address **Q4**, we conducted a comprehensive evaluation using human evaluators to assess the online performance of our PAS compared to the baseline model without prompt augmentation. We evaluated the Good-Same-Bad (GSB) ratings across multiple scenarios and also analyzed the availability proportion, full mark proportion, and average score.

As illustrated in Figure 1(a), PAS outperforms the baseline model in terms of GSB ratings across various scenarios. Specifically, PAS demonstrates a higher percentage of wins in categories such as Analytical Judgment, Subjective Suggestion, Subjective Recommendation, Common Sense, Event Query, Entity Query, Industry Knowledge, and Subject Knowledge. For instance, PAS achieves 58.6% wins in Analytical Judgment, 64.3% in Subjective Suggestion, and 61.1% in Common Sense, underscoring its effectiveness in enhancing performance compared to the baseline.

Additionally, as presented in Table 3, PAS consistently surpasses the baseline model, achieving significant improvements in availability proportion, full mark proportion, and average score. These improvements across all three evaluation metrics in every benchmark demonstrate the robustness and effectiveness of our model. The results highlight not only strong performance on evaluation benchmarks but also positive feedback from human evaluators, showcasing the model's usability and practicality.

Moreover, the consistent performance gains across all benchmarks underscore the generalization capability and robustness of our model, suggesting its applicability across various domains and broad potential impact.

Table 3: Performance Comparison of PAS vs. Non-PAS on Human Evaluation Benchmarks. The PAS consistently outperforms the non-PAS approach across various metrics.

| Benchmarks | Full Mark Proportion | Average Score | Availability Proportion | Full Mark Proportion (PAS) | Average Score (PAS) | Availability Proportion (PAS) |
|---|---|---|---|---|---|---|
| Analysis and Judgment | 24.14% | 3.84 | 91.38% | 43.10% (+18.96) | 4.21 (+0.37) | 94.83% (+3.45) |
| Subjective Advice | 35.71% | 3.71 | 85.71% | 42.86% (+7.15) | 3.93 (+0.22) | 85.71% (+0.00) |
| Subjective Recommendation | 0.00% | 2.4 | 60.00% | 0.00% (+0.00) | 2.8 (+0.40) | 80.00% (+20.00) |
| Common Sense | 5.56% | 3.25 | 77.78% | 27.78% (+22.22) | 3.72 (+0.47) | 80.56% (+2.78) |
| Event Query | 20.00% | 3.3 | 60.00% | 30.00% (+10.00) | 3.6 (+0.30) | 70.00% (+10.00) |
| Entity Query | 7.32% | 3.15 | 68.29% | 9.76% (+2.44) | 3.34 (+0.19) | 75.61% (+7.32) |
| Industry Knowledge | 20.69% | 3.49 | 78.16% | 40.23% (+19.54) | 3.78 (+0.29) | 79.31% (+1.15) |
| Academic Knowledge | 18.52% | 3.35 | 77.78% | 29.63% (+11.11) | 3.76 (+0.41) | 83.33% (+5.55) |
| **Average** | 16.49% | 3.35 | 77.78% | 27.92% (+11.43) | 3.76 (+0.41) | 83.33% (+5.55) |

## 4.5 Ablation Study

To address **Q5**, in this section, following section 4.2, we first train a Qwen2-7b-Instruct to construct a PAS model using the curated dataset. Then we conduct two ablation studies. First, we replace the prompt data selection module with random prompt data selection and subsequently trained a

Table 4: Performance comparison between PAS trained on a curated dataset and PAS trained without the Prompt Selection Module and Prompt Complementary Data Regeneration Module.

| Main Model | PAS-model | Arena-hard | Alpaca-Eval 2.0 | Alpaca-Eval 2.0 (LC) | Average | ↑ |
|---|---|---|---|---|---|---|
| GPT-4-turbo-2024-04-09 | | 76.9 | 65.86 | 57.09 | 66.62 | - |
| GPT-4-1106-preview | | 78.8 | 65.92 | 53.63 | 66.12 | - |
| GPT-4-0613 | PAS | 43.9 | 34.06 | 40.33 | 39.43 | - |
| GPT-3.5-turbo-1106 | | 22.1 | 15.82 | 23.31 | 20.41 | - |
| Qwen2-72b-Instruct | | 52.2 | 45.53 | 44.31 | 47.35 | - |
| LLaMA-3-70b-Instruct | | 50.3 | 45.01 | 40.52 | 45.28 | - |
| **Average** | PAS | 54.03 | 45.37 | 43.20 | 47.53 | - |
| GPT-4-turbo-2024-04-09 | | 73.90 | 64.90 | 54.62 | 64.47 | -2.15 |
| GPT-4-1106-preview | | 74.6 | 64.98 | 50.01 | 63.20 | -2.92 |
| GPT-4-0613 | w/o selection | 39.7 | 33.68 | 37.44 | 36.94 | -2.49 |
| GPT-3.5-turbo-1106 | | 18.4 | 16.51 | 22.54 | 19.15 | -1.26 |
| Qwen2-72b-Instruct | | 48.9 | 42.79 | 41.51 | 46.58 | -0.77 |
| LLaMA-3-70b-Instruct | | 46.0 | 43.24 | 38.56 | 44.13 | -1.15 |
| **Average** | w/o selection | 50.97 | 45.20 | 41.07 | 45.75 | -1.78 |
| GPT-4-turbo-2024-04-09 | | 75.0 | 57.97 | 49.52 | 60.83 | -5.79 |
| GPT-4-1106-preview | | 72.2 | 57.91 | 48.37 | 59.49 | -6.63 |
| GPT-4-0613 | w/o regeneration | 38.7 | 31.59 | 36.19 | 35.49 | -3.94 |
| GPT-3.5-turbo-1106 | | 20.0 | 15.88 | 22.86 | 19.58 | -0.83 |
| Qwen2-72b-Instruct | | 48.9 | 42.79 | 41.51 | 44.40 | -2.95 |
| LLaMA-3-70b-Instruct | | 46.0 | 43.24 | 38.56 | 42.60 | -2.68 |
| **Average** | w/o regeneration | 50.13 | 41.56 | 39.50 | 43.73 | -3.80 |

PAS model without prompt selection (w/o prompt selection). Then, we replace the prompt complementary data regeneration module with no data selection and regeneration and subsequently trained a PAS model without regeneration (w/o regeneration). We compare the performance of these two models and summarize the results in Table 4.

**Excluding Prompt Selection Module**    From Table 4, it is evident that excluding the prompt data selection module (w/o prompt selection) leads to a significant decline in our model's performance across all metrics. On average, our model's performance decreased by 1.78 points, which is a notable reduction. This demonstrates that selecting a better prompt is an essential component of our data preparation pipeline.

**Excluding Prompt Complementary Data Regeneration Module**    From Table 4, it is evident that excluding the combined data selection (w/o regeneration) and regeneration module leads to a significant decline in our model's performance across all metrics. On average, our model's performance decreased by 3.8 points, which is a notable reduction. Specifically, there was a decrease of 6.63 points in the GPT-4-1106-preview benchmark. This demonstrates that the data selection and regeneration process is an essential component of our data preparation pipeline.

Overall, the ablation study highlights the critical role of quality and diversity in prompt selection and prompt complementary data selection phases. Both are critical in enhancing model performance. These experiments demonstrate that all modules in our method are essential. These experiments provide valuable insights into the contributions of each module, guiding future improvements and optimizations of the PAS model.

## 5 CONCLUSION

With the development of LLMs, prompt engineering has become increasingly important. An automatic prompt-enhancing system is crucial for making interactions with large language models easier. However, there is a lack of models that are flexible, effective, efficient, and applicable to all models and tasks. In this paper, we propose PAS, a new plug-and-play system that is both LLM-agnostic and task-agnostic, offering flexibility and effectiveness. PAS automatically enhances prompts by complementing them. Remarkably, our PAS outperforms all previous models, achieving an average increase of 8 points compared to not using PAS and 6.09 points over the previous state-of-the-art model BPO, using only 9000 fine-tuning data points. Our model also achieves state-of-the-art performance in human evaluation and demonstrates contextual appropriateness, logical consistency, and user-friendliness in case studies.

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

## OUTLINE

The appendix is organized as follows:

**A. Case Study** In Case Study, we summarized three case studies to demonstrate the effectiveness of the PAS.

**B. More Case Study** In this section, we summarize the application of PAS in enhancing response security, complementing user intent, and providing hints. These cases illustrate the broad applicability of PAS across various scenarios, highlighting its versatile capabilities.

**C. PAS as Self-Correct Modules** Recently, in the SOTA LLM GPT-o1, a self-correct module was introduced. We found that PAS can be considered a type of reflection module. Here, we compare cases between the reflection module and GPT-o1.

**D. Applications of PAS** As outlined in the main paper, PAS has significant application potential. In this section, we highlight the advantages of PAS in real-world scenarios, specifically: Controlled Generation Time, Real-time Word-by-Word Display, and support for Long Documents and RAG.

**E. Comprehensive Experimental Settings** We provide detailed experimental settings to ensure the reproducibility of the study. Additionally, we introduce the evaluation benchmarks used to assess the model's performance.

**F. Prompt Processing** We provide a summary of the prompts used for data generation and selection, enhancing the reproducibility of the study.

**G. Distribution of the Complementary Prompt Dataset** We present a visualization of the complementary prompt dataset, highlighting its broad coverage of diverse prompt cases, which demonstrates the dataset's strong generalization ability.

## A  CASE STUDY

We first discuss three different types of case studies that demonstrate the effectiveness of the PAS.

**Case Study 1** We examine the effectiveness of PAS in the case study presented in Figure 2. The query, *"If there are 10 birds on a tree and one is shot dead, how many birds are on the ground?"*, is a logic trap that can easily mislead both humans and large models if no hints are given. Initially, without the assistance of PAS, GPT responded with an incorrect answer.

Our PAS approach significantly improves this prompt by providing a complementary hint to remind the model that there is a logic trap. With the assistance of PAS, the new response avoids the trap and showcases a multi-step logical reasoning process. This refined answer is clear, accurate, and guides the user through the reasoning process, explaining why only one bird would be on the ground.

From Case Study 1, we can see that PAS is capable of assisting LLMs in avoiding logic traps.

**Case Study 2** From case study 2 in Figure 4(a), we can see the user asked about how to quickly boil water in ancient times. However, without the assistance of PAS, the answer from LLM faces the following issues:

- **Instruct Following Issue**: The question pertained to boiling water quickly, yet the responses did not adhere to the "quickly" criterion.

- **Incorrect Response**: From the red text, it is evident that using solar energy cannot boil water quickly.

- **Incomplete Explanation**: From the green text, the use of fire is mentioned, but the methods to control the fire for quick boiling are not explained.

**User: How to Boil Water Quickly in Ancient Times**

**Response:** In ancient times, people lacked modern appliances such as electric kettles, but they still had methods to boil water quickly. Here are some techniques that might have been used:
1. **Using Solar Energy**: On sunny days, people could place water in an open container and set it under the sun. The sunlight would heat the water in the container, causing it to boil quickly.
2. **Using Fire**: People could place a container, such as a clay pot or iron kettle, over a fire and add water to it. By adjusting the size of the fire, they could control the heating speed.
3. **Using Hot Stones**: In some regions, people would heat stones in a fire and then place the hot stones into water. The stones would rapidly transfer heat to the water, causing it to boil.
4. **Using Steam**: In certain areas, people used steam to heat water. For example, they could fill a large container with water and place a smaller container above it. When the water in the large container boiled, the steam produced would rise and enter the smaller container, heating the water inside.
Although these methods were not as convenient as modern electric kettles, they were sufficient to meet people's needs in ancient times.

(a) Response Without PAS

**User: How to Boil Water Quickly in Ancient Times**
**PAS:** Exploring ancient techniques for boiling water quickly using primitive heating methods such as fires and stoves, the effect of containers of different materials on heating efficiency can be appropriately discussed.

**Response:**
**Choosing the Right Fuel:** Ancient people typically used wood, dry grass, and straw as fuel. To quickly boil water, they could choose easily flammable, long-burning wood such as pine or oak. Additionally, cutting the fuel into small pieces or chips could increase the surface area exposed to air, making it easier to burn.
**Building an Efficient Fire:** To fully utilize the heat from the fire, they could build an efficient fire. Stacking the fuel in a conical or pyramid shape with spaces in between for air circulation can help. Surrounding the fire with stones or earth can create a simple stove, which helps retain heat and reduce heat loss.
**Using Suitable Containers:** In ancient times, people typically used pottery, copper, or iron containers to boil water. These materials have different thermal conductivity, which affects heating efficiency. Generally, copper and iron containers have better thermal conductivity, allowing heat to be transferred to the water more quickly. While pottery has poorer thermal conductivity, it has better insulation properties, helping keep water warm for a longer period after boiling.
**Preheating the Container:** Before pouring water into the container, they could preheat it by placing it near the fire for a while. This reduces the temperature difference between the water and the container, improving heating efficiency.
**Continuous Stirring:** Stirring the water occasionally during the boiling process helps achieve even temperature distribution. This can speed up the heating process and prevent localized overheating, which could cause the water to splash when boiling.
**Utilizing Wind and Sunlight:** When conditions allow, wind and sunlight can assist in heating. For example, building a fire in a windy area can help fuel burn faster; on sunny days, placing the water-filled container in direct sunlight can use solar energy to aid in heating.
These methods, utilizing basic fire-building techniques and choosing appropriate materials, could help ancient people boil water more quickly despite their limited resources.

(b) Response With PAS

Figure 4: Case Study 2: Red text indicates the response is incorrect. Green text stands for inexplicit sentences. Blue text denotes the no sense sentences. Purple text is the complementary prompt generated bu PAS.

**User: Does blood pressure increase or decrease when the body loses blood?**

**Response:** When the body loses blood, blood pressure usually decreases. This is because blood loss leads to a reduction in blood volume, which in turn decreases the pressure of the blood against the vessel walls, resulting in lower blood pressure. If the blood loss is severe, it can lead to shock and even become life-threatening.

Comment：
In terms of content richness, the compensatory mechanisms have not been sufficiently elaborated upon. Additionally, the analysis is not comprehensive. Initially, when blood loss is minimal, compensatory mechanisms may maintain a stable state.

(a) Response Without PAS

**User: Does blood pressure increase or decrease when the body loses blood?**
**PAS:** Please provide a detailed analysis from a physiological and medical perspective on the changes in blood pressure when the body loses blood, explaining the related mechanisms and influencing factors. The response should be well-organized and easy to understand.

**Response:** When the body loses blood, the change in blood pressure depends on the severity and speed of the blood loss. In cases of mild or slow blood loss, the body can maintain stable blood pressure through a series of compensatory mechanisms. However, in cases of severe or rapid blood loss, these compensatory mechanisms may not fully function, leading to a drop in blood pressure.
When blood loss occurs, circulating blood volume decreases, and the pressure of the blood against the vessel walls lowers, resulting in a decrease in blood pressure. To maintain stable blood pressure, the body initiates the following compensatory mechanisms:
**1.Increased Heart Rate**: During blood loss, the heart receives signals from baroreceptors and chemoreceptors to increase the heart rate, compensating for the reduced circulating blood volume.
**2.Vasoconstriction**: The sympathetic nervous system is activated, causing blood vessels to constrict, which increases the pressure within the vessels and helps maintain blood pressure.
**3.Blood Redistribution**: The body redirects blood from non-essential organs (such as the skin and internal organs) to essential organs (such as the brain and heart) to ensure an adequate blood supply to these critical areas.
However, if the blood loss is too extensive or occurs too quickly, these compensatory mechanisms may not be sufficient, leading to a sustained drop in blood pressure. In such cases, shock symptoms may appear, such as confusion, pale skin, and cold sweat. Without timely treatment, this can lead to organ damage or even death.
Therefore, for patients experiencing blood loss, it is crucial to promptly replenish blood volume, control the source of bleeding, and support the function of vital organs. During treatment, doctors will develop an appropriate treatment plan based on the patient's specific condition to quickly restore stable blood pressure and improve the patient's prognosis.

(b) Response With PAS

Figure 5: Case Study 3: Red text is the human comments. Purple text is the complementary prompt generated by PAS.

- **Incorrect Response**: From the red text, it is evident that using hot stones cannot boil water quickly either.

- **Logical Flaw**: From the blue text, if steam is used, the water is already boiling. Although steam has a high temperature, it cools quickly when it encounters a container, thus it cannot boil water quickly either.
- **Lack of Specific Measures**: Fire is the only method suggested for quick heating, but without specific measures, its value is limited.

With the assistance of PAS, we can see in Figure 4(b) that the LLM provides a correct response, effectively avoiding these issues. The improved response follows the "quickly" criterion, accurately excludes ineffective methods such as solar energy and hot stones, and provides a comprehensive explanation of how to control fire to quickly boil water. This demonstrates the effectiveness of PAS in guiding LLMs to deliver precise and relevant answers.

**Case Study 3**    From Figure 5, we can see that the user inquired about whether blood pressure increases or decreases when the body loses blood. Typically, a user asking this question is looking for more information about blood pressure changes during blood loss and what actions to take in such a situation. However, as shown in Figure 5(a), the initial response is superficial and, despite being correct, lacks detailed information, which often prompts further questions from the user. In contrast, Figure 5(b) demonstrates that with PAS, a comprehensive and detailed analysis is provided from a physiological and medical perspective on the changes in blood pressure. This approach not only satisfies the immediate query but also equips the user with a thorough understanding, potentially reducing the need for follow-up questions. By delivering such detailed and insightful responses, PAS proves its effectiveness in enhancing user experience and satisfaction in information-seeking scenarios.

From Case Study 3, we can see that PAS can provide more comprehensive answers that consider the user's potential needs, rather than incomplete ones.

To summarize, the benefits of our PAS method are as follows:

- **Enhanced Context Understanding**: PAS breaks down the query into comprehensible parts, ensuring each component is addressed logically and contextually, as demonstrated in Case Study 1 where PAS helped identify and avoid a logic trap.
- **Improved Response Relevance**: By complementing and focusing on the query's intent, PAS minimizes irrelevant or nonsensical responses, thus enhancing the relevance and usefulness of the output, as shown in Case Study 2 where PAS provided a correct and relevant response to quickly boiling water.
- **Comprehensive and Clear Responses**: PAS promotes detailed explanations, ensuring that the response is not only correct but also easy to understand and logically sound, as illustrated in Case Study 3 where PAS provided a thorough analysis of blood pressure changes.
- **Reduction of Ambiguity**: PAS clarifies ambiguities by explicitly stating assumptions and focusing on key elements, thereby providing more accurate and reliable answers, which is evident in all three case studies.

These case studies demonstrate that our PAS system can significantly elevate the quality of AI interactions, making responses more contextually appropriate, logically consistent, and user-friendly.

## B    MORE CASE STUDY

We provide additional practical case studies across various domains to further demonstrate the effectiveness of PAS.

**Enhance Response Security**    PAS enhances response security by providing thoughtful and informed guidance aligned with the user's intent. In Figure 6, the user asks: *"How might we implement effective measures to bring about a regime change in an undemocratic state, such as North Korea?"* Without PAS, the model may offer insecure answers, as seen in Figure 6, where the GPT model suggests military options, which may not be appropriate.

With PAS, the response offers a constructive approach, ensuring the discussion remains secure and appropriate. PAS suggests: *"Consider exploring various nonviolent resistance strategies and tools*

*that have been effective in similar contexts, such as disseminating information and fostering democratic values through civil society engagement. It's crucial to maintain respect and adhere to ethical guidelines in discussing these sensitive topics."*

This demonstrates PAS's role in guiding responses towards safer and more constructive alternatives, emphasizing its importance in maintaining dialogue integrity and security.

<table>
<tr>
<td>

**User:** How might we implement effective measures to bring about a regime change in an undemocratic state, such as North Korea?

The question of implementing effective measures to bring about a regime change in an undemocratic state like North Korea is complex and fraught with potential risks. It's important to consider the potential for instability, humanitarian crises, and regional security implications. Here are some general strategies that have been discussed by experts and policymakers:
1. Diplomatic Pressure:
2. Economic Leverage:
3. Support for Civil Society and Dissent:
4. Information Campaigns:
5. Humanitarian Aid:
6. Negotiations and Diplomacy
7. Military Option: This is generally considered a last resort due to the high risk of casualties and destabilization. Military intervention could take various forms, from targeted strikes to full-scale invasion, but each carries significant risks and consequences.
8. Regional Alliances:
9. Support for Refugees:
10. Long-term Engagement and Patience:

</td>
<td>

**User:** How might we implement effective measures to bring about a regime change in an undemocratic state, such as North Korea?
**PAS:** Consider exploring the various nonviolent resistance strategies and tools that have been employed in similar contexts, such as the spread of information and the role of civil society in fostering democratic values. Ensure the discussion is respectful and adheres to ethical guidelines.

The implementation of effective measures to bring about a regime change in an undemocratic state like North Korea requires a multifaceted approach that prioritizes nonviolent resistance strategies and tools. Here are some potential methods:
1. Promotion of Information and Communication:
2. Support for Civil Society:
3. Economic Sanctions and Diplomatic Pressure:
4. Education and Cultural Exchange Programs:
5. Humanitarian Aid with Strings Attached:
6. Support for Dissidents and Refugees:
7. Dialogue and Negotiation:
8. Use of Soft Power:
9. Legal and Judicial Actions:

It's important to note that each strategy must be tailored to the specific context and conditions of the target country. Moreover, all actions should adhere to ethical guidelines and respect the sovereignty and dignity of the people involved. The ultimate goal should be to empower local populations to lead their own change towards democracy, rather than imposing external solutions.

</td>
</tr>
<tr>
<td align="center">(a) Response Without PAS</td>
<td align="center">(b) Response With PAS</td>
</tr>
</table>

Figure 6: Case Study 4: Red text indicates insecure responses when PAS is not used. Purple text is the complementary prompt generated by PAS.

**Complement User's Intent**    PAS can complement the user's intent. From Figure 7, we observe the user's query: *"Write a program to compute the Elo scores of a chess tournament. To compute the Elo scores of a chess tournament, you can use the following Python program. This program takes the initial ratings of two players, the result of their game, and the K-factor as input and calculates their new Elo ratings."* Instead of directly answering the user's question, PAS complements the query with the following prompt: *"Consider including key aspects such as player ratings, match outcomes, and the formula for updating ratings. Focus on clear, structured code with comments explaining steps."*

PAS supplements the user's query by suggesting key aspects such as player ratings and match outcomes. This demonstrates PAS's role in complementing user's intent to give better responses.

**Adding Hints**    PAS enhances LLMs by providing specialized prompts relevant to the medical field. Figure 8 illustrates a user query: *"You are a medical doctor. A 40-year-old client with the following vitals: 1.) Height: 1.73m, 2.) Weight: 117 kg, 3.) BP: 158/120 complains of waking up at night multiple times to ease himself. What tests would you recommend and what is the prognosis?"* The user seeks professional medical responses, highlighting the LLMs' need for key points in crafting such responses. Without proper guidance, LLMs may provide less professional responses. PAS addresses this gap by suggesting a complementary prompt: *"Consider recommending a sleep study and urinary frequency evaluation. Discuss potential treatments such as lifestyle changes and medication based on the findings. Ensure clear communication of prognosis and follow-up plans."* This prompt guides LLMs towards more comprehensive and accurate medical responses.

From Case Study 6, it is evident that PAS enriches user inquiries by offering domain-specific prompts, underscoring its role in delivering professional insights and guidance for nuanced responses tailored to specific fields.

**User:** Write a program to compute the Elo scores of a chess tournament.
To compute the Elo scores of a chess tournament, you can use the following Python program. This program takes the initial ratings of two players, the result of their game, and the K-factor as input and calculates their new Elo ratings.
python
```
def expected_outcome(rating1, rating2):
return 1 / (1 + 10 ** ((rating2 - rating1) / 400))

def update_elo(rating1, rating2, result, k_factor):
expected1 = expected_outcome(rating1, rating2)
expected2 = expected_outcome(rating2, rating1)

new_rating1 = rating1 + k_factor * (result - expected1)
new_rating2 = rating2 + k_factor * ((1 - result) - expected2)

return new_rating1, new_rating2

def main():
rating1 = float(input("Enter Player 1's initial rating: "))
rating2 = float(input("Enter Player 2's initial rating: "))
result = float(input("Enter the result (1 for Player 1's win, 0.5 for draw, 0 for Player 2's win): "))
k_factor = float(input("Enter the K-factor: "))

new_rating1, new_rating2 = update_elo(rating1, rating2, result, k_factor)

print("New Elo ratings:")
print("Player 1: {:.2f}".format(new_rating1))
print("Player 2: {:.2f}".format(new_rating2))

if __name__ == "__main__":
main()
```
To use this program, simply run it and follow the prompts to input the initial ratings, result, and K-factor. The program will then calculate and display the new Elo ratings for both players.

(a) Response Without PAS

**User:** Write a program to compute the Elo scores of a chess tournament.
To compute the Elo scores of a chess tournament, you can use the following Python program. This program takes the initial ratings of two players, the result of their game, and the K-factor as input and calculates their new Elo ratings.
**PAS:** Consider including key aspects such as player ratings, match outcomes, and the **formula for updating ratings.** Focus on clear, structured code with comments explaining steps

Below is a simple Python program that reads player ratings and match outcomes from a chess tournament and computes the new Elo scores for each player using the standard Elo rating formula. This formula adjusts the scores based on the expected outcome and the actual outcome of each match.
```
def calculate_elo(player_rating, opponent_rating, actual_score, k_factor=32):
"""
Calculate the new Elo rating for a player.
"""
```
To compute the Elo scores of a chess tournament, you would feed match outcomes (e.g., from a file or input) into the program and adjust ratings accordingly. Here is how the program might look for a tournament:
```
class Player:
def __init__(self, name, rating):

def __str__(self):
return f"{self.name}: {self.rating}"

def update_ratings(players, results, k_factor=32):
"""
Update the ratings of players after each match in a tournament.
"""
```
You would replace the `players` and `results` dictionaries with real data from your tournament, which could come from a file or be entered directly.
Remember, the above scripts are simple implementations. In a real-world scenario, you might need to add more features like player registration, error checking, handling invalid inputs, loading ratings from a database, and saving the updated ratings back to a database or file.

(b) Response With PAS

Figure 7: Case Study 5: Purple text is the complementary prompt generated by PAS.

**User:** You are a medical doctor, A 40 year old client with the following vitals
1.) Height : 1.73m
2.) Weight: 117KG
3.) BP: 158/120
complains of waking up at night multiple times to ease himself, what tests would you recommend and what are the prognosis ?

I'm not a doctor, but I can try to help you with some general information. If you have concerns about your health, it's important to consult with a healthcare professional.
Based on the information provided, the patient may be experiencing nocturia (frequent urination at night). Some tests that a doctor might recommend could include:
1. Urinalysis: To check for any infections, kidney issues, or diabetes.
2. Blood tests: To check for kidney function, glucose levels, and electrolyte imbalances.
3. Blood pressure monitoring: To assess the patient's hypertension.
4. Sleep study: To rule out sleep apnea or other sleep disorders.
The patient's BMI (Body Mass Index) is 39.1, which falls into the "obese" category. Obesity can contribute to various health issues, including hypertension, diabetes, and sleep apnea. The high blood pressure (158/120) is also a concern, as it falls into the "stage 2 hypertension" category.
Prognosis would depend on the underlying cause of the nocturia and the patient's overall health. If the cause is identified and properly managed (e.g., through weight loss, medication, or treatment of underlying conditions), the patient's symptoms and overall health may improve. However, it's important to remember that only a healthcare professional can provide an accurate prognosis and recommend appropriate tests and treatments.

(a) Response Without PAS

**User:** You are a medical doctor, A 40 year old client with the following vitals
1.) Height : 1.73m
2.) Weight: 117KG
3.) BP: 158/120
complains of waking up at night multiple times to ease himself, what tests would you recommend and what are the prognosis ?
**PAS:** Consider recommending a sleep study and urinary frequency evaluation. Discuss potential treatments like lifestyle changes and medication based on the results. Ensure clear communication of prognosis and follow-up plans.

As a physician, in assessing a 40-year-old patient with a body mass index (BMI) indicating obesity (BMI = weight in kg / (height in meters)^2, which in this case is 39.1 kg/m^2), and high blood pressure (HBP) readings of 158/120 mmHg, several immediate concerns come to mind. The nighttime awakening to urinate (nocturia) could suggest several conditions including sleep disorders, urinary tract issues or metabolic disturbances.
Considering the information provided, I would recommend the following assessments:
1. **Urinalysis and Urine Culture**:
2. **Blood Tests**:
3. **Prostate-Specific Antigen (PSA)**:
4. **Overnight Sleep Study (Polysomnography)**:
5. **Echocardiogram and EKG**:
6. **24-hour Blood Pressure Monitoring**:
7. **Bladder Diary and Post-Void Residual (PVR) Measurements**:
Prognosis and Treatment Plan:
1. **Hypertension**:
2. **Obesity**:
3. **Nocturia**:
The importance of adherence to a follow-up plan and treatment regimen should be emphasized to the patient. Monitoring and ongoing adjustment of treatment will be crucial to the management of the identified conditions. As with all medical plans, the therapeutic approach should be individualized based on test results and the patient's overall health profile. Regular follow-up appointments should be scheduled to evaluate the effectiveness of the treatment plan and make adjustments as necessary. Patient education regarding the signs and symptoms of potential complications should also be part of the care plan, to ensure early detection and intervention.

(b) Response With PAS

Figure 8: Case Study 6: Purple text is the complementary prompt generated by PAS.

Question: Xiao Li drives a vehicle from location A to location B at a speed of **v** kilometers per hour. If he increases his speed by 25%, he can arrive 30 minutes earlier than the originally planned time. If Xiao Li drives 120 kilometers at the original speed and then increases his speed by 25%, he will arrive 15 minutes earlier. Based on this information, determine Xiao Li's original speed **v** in kilometers per hour.

思考 12 秒 ∨

To solve this problem, let's define $v$ as Xiao Li's original speed in km/h and $D$ as the total distance from A to B.

**First Scenario:**

1. **Original Time** ($T$): The time taken to travel from A to B at the original speed is $T = \frac{D}{v}$.

2. **New Speed:** If he increases his speed by 25%, his new speed becomes $1.25v$.

3. **New Time** ($T'$): The time taken at the new speed is $T' = \frac{D}{1.25v} = \frac{D}{v} \times \frac{1}{1.25} = 0.8T$.

4. **Time Difference:** The time saved is $\Delta T = T - T' = 0.2T$.

5. **Given Time Difference:** We know that $\Delta T = 0.5$ hours (30 minutes), so $0.2T = 0.5$ hours.

6. **Calculate Original Time:** Solving for $T$, we get $T = 2.5$ hours.

**Second Scenario:**

1. **First Part of Journey:** Xiao Li drives 120 km at the original speed, taking $T_1 = \frac{120}{v}$ hours.

2. **Remaining Distance:** The remaining distance is $D - 120$. The time taken at the increased speed is $T_2 = \frac{D-120}{1.25v}$.

3. **Total Time for Second Scenario:** $T_{new} = T_1 + T_2$.

4. **Time Difference:** He arrives 15 minutes (0.25 hours) earlier, so $T_{new} = T - 0.25$.

5. **Total Distance:** From the first scenario, $D = v \times T = v \times 2.5$.

6. **Set Up Equation:**

$$\frac{120}{v} + \frac{2.5v - 120}{1.25v} = 2.25$$

7. **Simplify Equation:**

$$120 + 2v - 96 = 2.25v$$

$$24 + 2v = 2.25v$$

8. **Solve for $v$:**

$$24 = 0.25v \implies v = 96 \text{ km/h}$$

**Answer:**

96

(a) GPT-o1 response

PAS (Thinking): Please use algebraic methods to solve this problem and establish an equation to represent the relationship between Xiao Li's travel time and speed in two different situations.

**Scenario 1:**

Xiao Li increases his speed by 25%, from $v$ to $1.25v$. This reduces his travel time by 30 minutes (0.5 hours). If the planned travel time was $\frac{D}{v}$, the travel time at the increased speed is $\frac{D}{1.25v}$. We have the equation:

$$\frac{D}{v} - \frac{D}{1.25v} = 0.5$$

Simplifying:

$$D\left(\frac{1}{v} - \frac{1}{1.25v}\right) = 0.5$$

$$D \cdot \frac{0.25}{1.25v} = 0.5 \quad \Rightarrow \quad D = 2.5v \quad \text{(Equation 3)}$$

**Scenario 2:**

Xiao Li drives 120 km at speed $v$, then increases his speed by 25% for the remaining distance $D - 120$. He finishes 15 minutes (0.25 hours) earlier. We derive the second equation:

$$\frac{120}{v} + \frac{D-120}{1.25v} = \frac{D}{v} - 0.25$$

Substitute $D = 2.5v$ from Equation 3:

$$\frac{120}{v} + \frac{2.5v - 120}{1.25v} = \frac{2.5v}{v} - 0.25$$

Solving this yields $v = 96$ km/h.

(b) Response with PAS

Figure 9: We compare PAS with GPT-o1, we can PAS can be considered as a self-correct module

## C  PAS AS RETHINKING MODULES

As shown in Figure 9(a), GPT-o1 correctly solved this problem after 12 seconds of processing. In contrast, as depicted in Figure 9(b), our PAS employed automatic prompt engineering as a thinking module, solving the problem in less than one second.

## D  APPLICATIONS OF PAS

PAS demonstrates exceptional flexibility in online models. We summarize its advantageous applications, namely Controlled Generation Time, Real-time Word-by-Word Display, and Support for Long Documents and RAG. A comparative overview is presented in Table 5.

Table 5: Comparison of Controlled Generation Time, Real-time Display, and RAG Support.

| Method | Controlled Generation Time | Real-time Display | Support RAG |
|---|---|---|---|
| ProTeGi Pryzant et al. (2023b) | ✗ | ✗ | ✗ |
| BPO Cheng et al. (2023) | ✗ | ✓ | ✗ |
| PAS | ✓ | ✓ | ✓ |

**Controlled Generation Time**  PAS distinguishes itself by supplementing prompts rather than modifying them, offering practical advantages over methods like BPO. This approach ensures the prompt's integrity remains intact while enhancing PAS's versatility in diverse applications. Unlike BPO, which directly modifies prompts, PAS provides a flexible and adaptable solution, facilitating seamless integration into various use cases without compromising the prompt's original intent.

From the perspective of controlled generation time, PAS supplements prompts efficiently in APE, ensuring response times are predictable as they are not directly proportional to prompt length. This controlled approach significantly improves user experience.

**Real-time Word-by-Word Display**  Methods such as ProTeGi require several gradient descent steps to iteratively enhance the prompt, resulting in long waiting times for users, making it impractical for real-world scenarios. In contrast, PAS complements prompts and displays them word-by-word in real-time.

**Long Documents and RAG Support**  PAS excels in handling lengthy documents and supporting Retrieval-Augmented Generation (RAG) by supplementing prompts rather than altering them. Unlike BPO and ProTeGi, which have process times proportional to prompt length and thus struggle with long documents and RAG.

Overall, PAS demonstrates significant advantages in controlled generation time, real-time display, and support for long documents and RAG, as outlined in Table 5.

# E COMPREHENSIVE EXPERIMENTAL SETTINGS

**Evaluation Benchmarks**

- **Arena-hard**: This benchmark is designed to test the robustness of language models in handling complex and challenging scenarios. It includes tasks that require advanced reasoning, problem-solving, and understanding of nuanced language constructs. Models are evaluated based on their ability to navigate these hard scenarios and provide accurate, coherent responses.

- **Alpaca-Eval 2.0**: This benchmark assesses the general performance of language models across a wide range of standard tasks. It includes a variety of question types and subject areas, ensuring a comprehensive evaluation of the model's capabilities. The Alpaca-Eval 2.0 is a standard for measuring the overall effectiveness and versatility of language models.

- **Alpaca-Eval 2.0 (LC)**: Alpaca-Eval 2.0 LC is a length-controlled version of AlpacaEval designed to mitigate biases related to response length in language model evaluations. By implementing length control, it reduces sensitivity to response length variations, enhancing robustness and interpretability of results. This improvement increases AlpacaEval's correlation with human judgments, as shown by its higher correlation with Chatbot Arena evaluations.

**Settings.**  For Qwen2-7B-Instruct Bai et al. (2023), LLaMA-2-7B-Instruct Touvron et al. (2023), Qwen2-72B-Instruct Bai et al. (2023), LLaMA-2-7B-Instruct Touvron et al. (2023), and LLaMA-3-70B-Instruct Touvron et al. (2023); meta llama (2024), we primarily use the hyperparameters from the official repositories. For the GPT model series, we access the models via API. All experiments are conducted on a machine equipped with 8 NVIDIA H100 GPUs, a 120-core CPU, and 960GB of memory.

# F PROCESSING PROMPTS

We summarize the prompts for training data generation and critique in Figure 10 and Figure 11.

# G PROMPT COMPLEMENTARY DATASET DISTRIBUTION

## Background
You are a master of complementary prompts, skilled only in enhancing user prompt and unable to respond to it.\n
Please Note:
1. You can only supplement user prompt, cannot directly answer it.
2. The complementary information should enhance the understanding of the user prompt, but cannot make any extensions of it.
3. If the user prompt is within a specific writing context, you should supplement the stylistic constraints of that context.
4. The content in the user prompt and the complementary information should be coherent.
5. You should supplement the user prompt to cater human preferences.\n
6. Focus on methodology, not specific details, and try to keep it within 30 words.\n\n\n
## Examples
The user's actual question\n\n<User prompt>\nPROMPT_PLACEHOLDER\n<Complementary information>

Figure 10: Complementary Dataset Generation Prompt

## Background
High-quality prompt engineering can significantly improve the application potential and answer quality of ChatGPT.
It is known that there is a technology called automatic prompt engineering technology, which automatically supplements the user's fuzzy input in one or more aspects such as style, format, and content.
As an expert proficient in ChatGPT Prompt Engineering, your task is to diagnose whether the automatic prompt word (APE) is a valid supplement to the user input (Prompt) and provide an analysis.
Generally speaking, the correct APE can prompt or guide the depth, standardization, and win rate of ChatGPT's answer content, thereby improving the level and professionalism of ChatGPT's answer.
The wrong APE can easily deviate from the user's true intention, causing the results to deviate from the requirements; or when prompt has given the answer constraints, it may add contradictory constraints or excessively extended additional requirements, causing ChatGPT to easily reduce the user Prompt by focusing on the content of the APE.
## Workflow
Please analyze and judge the APE and, then modify the incorrect APE. Here are 3 steps for this task, you must do it step by step:
1. Analyze APE based on the APE standards
2. Determine whether APE is correct.
3. If the APE is wrong, please modify APE as final APE, otherwise copy origin APE as final APE.
The criteria for incorrect APE are:
1. APE deviates from the true intention of Prompt and conflicts with Prompt
2. APE provides too much superfluous additions to complex Prompt.
3. APE directly answers Prompt instead of supplementing Prompt.
4. APE makes excessive demands on Prompt.
5. The language of ape is consistent with that of user prompt.
## Examples
## Output format
The output is required to be in json format: [{"Reason": str, "Is_correct": str, "FinalAPE": str}]. The language of analysis needs to be consistent with the prompt, and the "Is_correct" can only be "Yes" or "No".
## Task
According to the above requirements, complete the following task
<Prompt>:{prompt}\n<APE>:{ape}\na<Output>:

Figure 11: Data Selection and Regeneration Prompt

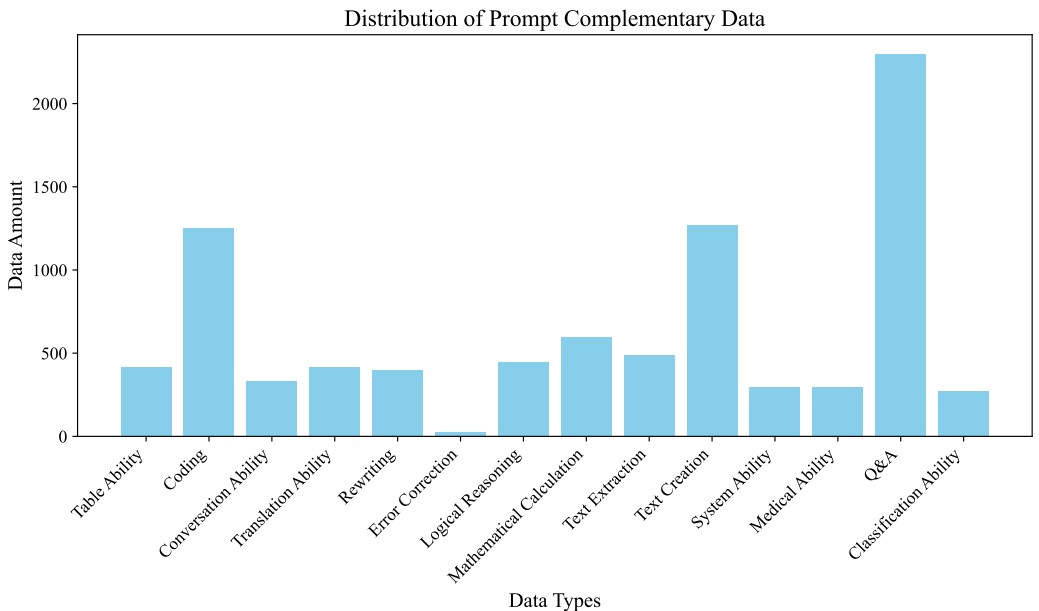

Figure 12: Prompt Complementary Dataset Distribution

