# OpenReview forum: "PAS: Plug-and-Play Prompt Augmentation System"
_ICLR.cc/2025/Conference — ICLR 2025 Conference Withdrawn Submission_

### Official Review · Reviewer_jzfZ · 2024-10-28

**Soundness:** 3
**Presentation:** 2
**Contribution:** 3
**Rating:** 5
**Confidence:** 5

**Summary:**

This paper presents a "plug-and-play" system called PAS for optimizing user prompts. The PAS system consists of two components: dataset construction and model fine-tuning. In the dataset construction phase, PAS first performs deduplication, selection, and classification on the data; it then leverages the few-shot learning capabilities of large language models to generate complementary prompts for fine-tuning data, followed by an evaluation to select high-quality data. In the model fine-tuning phase, the model is fine-tuned using the generated <prompt, complementary prompt> data pairs. Experimental results show that PAS outperforms BPO in terms of performance, validating the effectiveness of PAS.

**Strengths:**

1. The authors propose a plug-and-play system that does not require continuous iteration to improve prompts, unlike other automatic prompt optimization methods.

2. The experimental section includes a detailed discussion of various large language models.

**Weaknesses:**

1. The baseline methods in the experimental section only include BPO, which is insufficient to demonstrate PAS as the current state-of-the-art (SOTA) solely based on its performance exceeding BPO. How does PAS's generated prompts compare with iterative methods like APO, OPRO, and PromptAgent? I have concerns regarding this. Based on my experience, adding supplementary descriptions to prompts can indeed enhance the performance of large language models, but it is challenging to surpass methods like APO, OPRO and PromptAgent, which involve comprehensive prompt rewriting. I suggest that the authors add comparative experiments with these baselines and conduct comprehensive comparisons on more diverse datasets, such as mathematical problems (GSM8K), commonsense questions (CommonsenseQA), and language understanding tasks (SST-5, TREC), among others.

2. PAS generates complementary prompts using large language models and filters them through the model as well. However, previous research has shown that large language models may exhibit self-enhancement bias, meaning LLMs may have a tendency to favor their own generated answers. The authors' approach may need to address this potential issue, such as using another model to evaluate the quality. Additionally, past research has shown that the improvement achieved by prompts for large language models can sometimes be inexplicable. For example, OPRO found that on the GSM8K dataset, the prompt “Take a deep breath and work on this problem step-by-step.” performs nearly 9% better than “Let's think step by step.” Therefore, relying solely on LLM-based data selection from a semantic perspective may be insufficient. The authors should consider a more comprehensive evaluation approach, such as generating results using prompt + complementary prompt on the LMSYS-1M and WildChat datasets and directly evaluating with corresponding metrics.

3. In Appendix D, the author mentions that PAS is considered Controlled Generation Time. How is this specifically demonstrated? BPO also inputs prompts into the LLM and receives optimized prompts in return, so why isn’t BPO considered Controlled Generation Time? Additionally, the author claims that PAS supports Long Documents and RAG. How is this reflected, and why doesn’t BPO qualify?
I feel that the author’s analysis does not clearly address these issues. For these two points, I recommend that the author add more compelling evidence, such as additional experiments or a more detailed comparative analysis.

4. Writing suggestions:

a) The reference for BPO in line 94 should be added earlier; the citation appearing only by line 106 might impact reading fluency.

b) In the related work section, the first paragraph of the “Automatic Prompt Engineering” section addresses prompt design techniques but has little relevance to the “Automatic” aspect of APE. I recommend reclassifying it into a different related work section (or removing it), such as “Prompt Design” or “Prompt Engineering,” while keeping the other parts under “Automatic Prompt Optimization.” I hope the authors will consider this suggestion.

c) The titles of sections 3.2 and 3.3 (“PROMPTS COMPLEMENTARY DATASET” and “PROMPT COMPLEMENTARY DATASET”) are too similar, making it challenging for readers to distinguish between them. Adjusting them for clarity would be helpful.

Based on the questions I raised in the "Questions" section, I believe that this paper is generally unclear in its presentation, I strongly suggest that the author provides a clearer explanation in the next version.

**Questions:**

1. Regarding the "Quality Selection" in Section 3.1, I am unclear about how the authors derived prompt scores using the Baichuan 13B model. Could you provide a more detailed explanation of this process?

2. For the "Classification" in Section 3.1, why did the authors not directly use GPT-3.5-turbo or GPT-4 with in-context learning to complete the classification task? Fine-tuning Baichuan 13B with 60,000 data entries for classification seems somewhat redundant.

3. In the "Prompt Complementary Dataset Generation" section, the authors repeatedly emphasize the use of few-shot learning. Could examples be provided for the “Data Generation” and “Data Selection and Regeneration” parts? Additionally, which model does PAS use in this section?

4. It appears that the authors did not conduct a separate experiment to address Q3. Was this an oversight in the writing?

5. I am not familiar with the details of benchmarks, and the paper lacks the corresponding citations or links. Additionally, what are the evaluation metrics for these benchmarks? How many data samples are included in the author’s test set? I hope the authors can provide some clarification.

---

### Official Review · Reviewer_G9Uc · 2024-10-31

**Soundness:** 2
**Presentation:** 1
**Contribution:** 4
**Rating:** 5
**Confidence:** 4

**Summary:**

The paper presents PAS, an LLM-based system designed to simplify and enhance prompt engineering for LLMs. Traditional prompt engineering can be challenging and time-intensive, so PAS leverages automatically generated complementary datasets to provide an efficient, flexible solution that augments user prompts without requiring extensive user input or model retraining. The system outperforms previous automatic prompt engineering models by achieving higher performance with only 9,000 data points, showing an average improvement of 6.09 points over its predecessor. PAS can be integrated with any LLM, offering a versatile, user-friendly approach to improve prompt effectiveness across diverse tasks while also excelling in human evaluations, demonstrating high usability and broad application potential.

**Strengths:**

1. **Novelty**: The authors claim their work is the first to curate a prompt complementary dataset without human labor.
2. **Technical Implementation**: Proves a pipeline and the necessary code to reproduce the results.
3. **Contribution**: The paper tackles a common problem in using LLM, prompt engineering. The work is well-motivated and its contribution is helpful and useful for the community.

**Weaknesses:**

1. **Poor Citation Practice**: The paper uses several important benchmarks for its main evaluation results, yet did not cite their works (eg. Arena-Hard, AlpacaEval). The authors need to make sure to credit all the works they used from other researchers.

Suggestion: The authors should add the appropriate citations and update the reference list.

2. **Clarity**: The paper has clarity issues, including several important parts of the paper being confusing. Examples and comments below:

> Line 262: We manually design 4-5 few shot examples for each category, where we call it few-shot golden data. Then we utilize the prompt dataset from section 3.1 to generate high-quality (prompt, complementary prompt) pairs utilizing few-shot learning.

It is unclear how or under what criteria the few shot examples are designed. They author did not link to these few-shot examples nor explain what necessary steps were taken to ensure these design choices are unbiased.

> Section 4.4: Human Evaluation

The paper did not explain how one of the main evaluation results, human evaluation, was conducted. Not knowing how the experiment was conducted (eg. who were the participants in evaluation and what specific dataset used) raises questions about the validity of the experiment.

Suggestion: Include information such as the number and qualifications of evaluators, the evaluation criteria provided to them, the dataset used, and any measures taken to ensure unbiased assessments.

3. **Lack of validations**: The presented method suffers from lack of validations, especially in crucial components which leverage LLM for decision making, which are known to be unreliable. Examples and comments below:

> Line 244: Here $Q_{score}(p_i)$ represents the quality score assigned by BaiChuan 13b model to prompt $pi_i$, and $\tau$ denotes the quality threshold. By employing quality selection, we aim to enhance the overall quality of the prompt data.

> Line 251: This results in a classification model capable of categorizing prompts into common categories such as Q&A and coding.

The authors claim the curation pipeline selects high quality data and accurately classifies categories. However, the author did not provide any validation results. Presentation of quantitative metrics for data quality selection and classification accuracy would be beneficial.

3. **Generalizability**
The authors claim their system can be generalized to any LLMs and obtain SOTA performance. However, this claim raises concerns. Examples and comments below:

> Line 392: To address Q1, we used Qwen-2-7B-Instruct as the base model due to its outstanding performance.

The author chose to present Qwen-2-7B-Instruct’s results because they are more favorable. To demonstrate the method is indeed robust to model as the author claimed, results should be shown for more models.

Suggestion: The authors should include a comprehensive table or figure showing results across a wider range of models, including both smaller and larger models from different families. This would help demonstrate the claimed robustness and generalizability.

> Line 386: To evaluate the effectiveness of our PAS model, we used three comprehensive benchmarks Arena-hard, Alpaca-Eval 2.0 and Alpaca-Eval 2.0 (LC) to thoroughly assess the model’s performance.

These benchmarks may not be representative of the real-world use case of LLM. The authors claim they are comprehensive, however, Arena-Hard contains difficult user queries from Chatbot Arena and AlpacaEval are mainly information retrieval tasks. For instance, Arena-Hard queries are selected for being well-defined, and do not necessarily require complementary augmentation. Unfiltered prompts in the wild, which are not well-defined, might suit better for evaluating the effectiveness of PAS. Hence, it is difficult to tell from these results whether PAS truly improves compared to the baseline.

Suggestion: The authors should include an additional evaluation using a dataset of unfiltered, real-world prompts. Discussion the limitations of the current evaluation benchmarks and how to address these limitations in future work would also be beneficial.

4. **Numerical Significance**

> Line 439: Notably, PAS exhibits a marked improvement in performance metrics compared to BPO, exceeding the baseline by 3.41 points on average. This is particularly evident in models like GPT4-0613, where the average score improvement is as high as 5.89 points. Even in cases where the improvement is smaller, such as Llama3-70b-Instruct and GPT-4-turbo-2024-04-09, PAS still manages to outperform BPO for more than 1 point, indicating its robustness and consistency.

It is difficult to tell whether the improvement is truly significant, when the difference between PAS and baseline is within variance of the benchmark. The author should provide confidence intervals for the presented results. It also appears that Arena-Hard provides confidence interval for model evaluation, which the authors should report.

**Questions:**

1. How was the Human Evaluation (Q4) conducted? What was the dataset chosen and who were the evaluators?

2. The chosen baseline BTO is relatively old. Are there other existing improved systems which could better serve as baselines?

3. How were the few-shot examples designed? What was the motivation behind the design choices? What was the accuracy of the categorization? Also what was the accuracy of the classification model?

4. The authors claim their model is user-friendly. Could the author shed light on what was conducted to demonstrate this, since the humane valuation benchmark does not include user-friendliness as a metric?

5. What was the reason to not compare PAS against BPO on the human evaluation benchmark?

---

### Official Review · Reviewer_QZNF · 2024-11-05

**Soundness:** 2
**Presentation:** 2
**Contribution:** 2
**Rating:** 3
**Confidence:** 4

**Summary:**

This paper introduces PAS, a Plug-and-Play Prompt Augmentation System designed to enhance the performance of Large Language Models (LLMs) through automatic prompt engineering. PAS addresses the challenges of existing APE methods (poor effectiveness, low flexibility, and limited practicality) by employing a two-stage alignment paradigm: a Prompt Augmentation (PA) model generates complementary content tailored to the original prompt, which is then fed into the LLM.  The PA model is trained on a curated dataset of prompt-complementary prompt pairs, created without human labor. Experiments show PAS achieves state-of-the-art performance on multiple benchmarks, outperforming previous SOTA model BPO.

**Strengths:**

Novelty: The paper presents a unique approach to APE by focusing on prompt complementation rather than rewriting. This potentially preserves the user's original intent, a key advantage over rewriting-based methods like BPO. The automatic generation of a complementary prompt dataset without human labeling is interesting.

Strong Empirical Results: The experimental results demonstrate significant performance gains over the baseline and the previous SOTA, BPO, across a range of LLMs and benchmarks. The average improvement of 8 points over the baseline and 6.09 points over BPO is substantial.

Efficiency:  PAS achieves SOTA performance with a relatively small training dataset (9000 data points), highlighting its efficiency.  The focus on controlled generation time also contributes to efficiency and makes real-time application feasible.

Flexibility and Applicability: PAS is designed to be plug-and-play, compatible with various LLMs and applicable to different tasks. The demonstrated applicability to long documents and RAG showcases its broad potential.

Human Evaluation: The inclusion of human evaluation reinforces the practical benefits of PAS. The superior GSB ratings and improvement across evaluation metrics (availability, full mark proportion, average score) suggest improved user experience.
Ablation Study: The ablation study provides insights into the importance of each module in the PAS pipeline, specifically prompt selection and data regeneration. This strengthens the paper's claims about the effectiveness of the proposed methodology.

**Weaknesses:**

Limited Details on Dataset Creation: While the paper describes the process of creating the prompt-complementary prompt dataset, more details are needed. Specifically, the types of prompts included in the initial dataset (LMSYS-1M and WildChat), the criteria for quality selection and classification categories should be further elaborated. A more detailed analysis of the dataset distribution (Figure 12) with examples would be beneficial.

Clarity on PA Model Architecture: The paper lacks specifics about the architecture and training details of the PA model. Information on the LLM used for training the PA model, the fine-tuning process (SFT), and hyperparameter choices is essential for reproducibility and understanding the system's inner workings.

Comparison with other Methods: The paper primarily focuses on comparison with BPO.  Including comparison with a wider range of APE methods (OPRO, APO, APE) would strengthen the paper's claims of achieving SOTA performance. Also there are other prompt augmentation methods which could/should be studied.

Discussion on Limitations: The paper could benefit from a more explicit discussion of the limitations of PAS. For instance, potential failure cases where the PA model fails to generate relevant complementary content, or the impact of the quality of the base prompt on the effectiveness of PAS, could be addressed.

**Questions:**

What is the specific architecture and training procedure of the PA model? What LLM was used for fine-tuning, and what were the hyperparameter choices?
Can you provide more details on the diversity and characteristics of the prompts in the training dataset? Can you share some examples from different categories shown in Figure 12?
How does PAS handle very long or complex prompts where generating relevant complementary content might be challenging?
What are the computational costs associated with using PAS, and how do they scale with the length of the prompt?
Have you considered any alternative approaches for generating complementary prompts, such as using retrieval methods or employing a multi-stage generation process?
What are the future directions for improving PAS, such as incorporating user feedback or exploring different types of complementary content?

---

### Official Review · Reviewer_wsCv · 2024-11-05

**Soundness:** 2
**Presentation:** 2
**Contribution:** 2
**Rating:** 3
**Confidence:** 4

**Summary:**

This paper presents a plug-and-play system called the Prompt Augmentation System (PAS) for generating prompts tailored to various tasks. PAS first curates a complementary prompt dataset for different categories and then uses this dataset to train an LLM for complementary prompt generation. During inference, the original prompt and the complementary prompt are used together. The authors demonstrate the effectiveness of their system compared to BPO on various datasets, along with human evaluations.

**Strengths:**

Curation of prompt dataset for fine-tuning the model is quite interesting

**Weaknesses:**

1.	I agree that prompting is central to improving LLM performance, and this work is timely and interesting. However, my first concern is the positioning of the work relative to the rich literature on automated prompt engineering techniques. While the authors briefly describe other approaches, they do not clearly articulate how PAS is different from and better than these techniques.

2.	My second concern is that the paper fails to clearly articulate the novelty of the work. While generating complementary prompts is interesting, there is nothing new in this approach. Additionally, once the data is generated, a simple supervised fine-tuning (SFT) is applied for complementary prompt generation.

3.	My third concern is that the paper lacks a comprehensive comparison to various other approaches (baselines). The authors only compare their method against BPO, but it’s unclear why they chose this single baseline, especially when there are several recent approaches, such as OPRO, EVOPrompt, PromptBreeder, and PromptWizard.

**Questions:**

1.	The authors fail to position their work within the rich literature on automated prompting techniques. For example, they do not discuss how their fine-tuning of LLMs with a prompt dataset is similar to or different from recent soft-prompting approaches, such as InstructZero and Use Your Instinct.

2.	In section 2.1 (lines 166–170), the authors dismiss evolutionary algorithms like EvoPrompt and PromptBreeder, citing challenges in practical applications. However, it’s unclear why PAS is preferable to these approaches, especially given that PAS also relies on prompt data and uses examples/datasets for scoring the prompts. What, specifically, are the challenges with other approaches?

3.	Clearly positioning the work in the literature is critical. The authors should justify why current approaches, particularly those mentioned above, fall short, and how PAS addresses these gaps. This explanation is completely missing from the current paper.

4.	The key contributions and novelty of the proposed approach are unclear. What exactly is novel in the approach? If it’s the prompt dataset generation, I don’t see any innovative techniques or insights developed here. In fact, the dataset appears highly restrictive to the categories and prompts from previous datasets rather than new tasks, especially in real-world settings. It’s unclear how well the prompts adapt to new categories, scenarios, etc.

5.	Why compare with BPO, an older approach from 2023, when several new techniques have emerged in the past six months, including EvoPrompt, PromptBreeder, PromptWizard, InstructZero, and Use Your Instinct?

6.	I strongly suggest that the authors compare their approach with state-of-the-art methods like EvoPrompt, PromptBreeder, PromptWizard, InstructZero, and Use Your Instinct. Additionally, I recommend expanding the datasets to include diverse tasks, such as Big Bench Hard, Math, or other domain-specific datasets. With the current evaluations, it is challenging to gauge the true performance of the proposed approach.

7.	The authors should present a cost analysis for PAS, detailing the time, tokens, and API calls required to generate a new prompt and comparing these metrics with other approaches.

8.	The authors report results from human evaluators, but it’s unclear how many evaluators were used, what instructions they received, and the level of inter-user agreement. Please provide these details to aid understanding. Additionally, the metrics in Table 3 are undefined, making it difficult to interpret the results.

---

### Note · Authors · 2024-11-21

I have read and agree with the venue's withdrawal policy on behalf of myself and my co-authors.